# Nebulized Non-Immunogenic Staphylokinase in the Mice Acute Lung Injury Model

**DOI:** 10.3390/ijms23169307

**Published:** 2022-08-18

**Authors:** Sergey S. Markin, Roman D. Lapshin, Olga S. Baskina, Svetlana A. Korotchenko, Irina V. Mukhina, Sergei V. Ivanov, Mikhail P. Semenov, Valerii V. Beregovykh, Andrey M. Semenov

**Affiliations:** 1Experimental Drug Research and Production Zone, Institute of Biomedical Chemistry, 119121 Moscow, Russia; 2LLC “SuperGene”, 119270 Moscow, Russia; 3Central Research Laboratory, Privolzhsky Research Medical University, 603005 Nizhny Novgorod, Russia

**Keywords:** acute lung injury, mice model, fibrin-fibrinogen deposit, nebulized non-immunogenic staphylokinase, fibrimolysis

## Abstract

Acute lung injury (ALI) as a model of acute respiratory distress syndrome is characterized by inflammation, complex coagulation, and hematologic abnormalities which result in the formation of fibrin-platelet microthrombi in the pulmonary vessels with the rapid development of progressive respiratory dysfunction. We hypothesize that a nebulized fibrinolytic agent, non-immunogenic staphylokinase (nSta), may be useful for ALI therapy. First, the effect of the nebulized nSta (0.2 mg/kg, 1.0 mg/kg, or 2.0 mg/kg) on the coagulogram parameters was studied in healthy rats. ALI was induced in mice by nebulized administration of lipopolysaccharide (LPS) at a dose of 10 mg/kg. nSta (0.2 mg/kg, 0.4 mg/kg or 0.6 mg/kg) was nebulized 30 min, 24 h, and 48 h after LPS administration. The level of pro-inflammatory cytokines was determined in the blood on the 8th day after LPS and nSta administration. The assessment of lung damage was based on their weighing and microscopic analysis. Fibrin/fibrinogen deposition in the lungs was determined by immunohistochemistry. After nSta nebulization in healthy rats, the fibrinogen blood level as well as activated partial thromboplastin time and prothrombin time did not change. In the nebulized ALI model, the mice showed an increase in lung weight due to their edema and rising fibrin deposition. An imbalance of proinflammatory cytokines was also found. Forty percent of mice with ALI without nSta nebulization had died. Nebulized nSta at a dose of 0.2 mg/kg reduced the severity of ALI: a decrease in interstitial edema and inflammatory infiltration was noted. At a dose of 0.4 mg/kg of nebulized nSta, the animals showed no peribronchial edema and the bronchi had an open clear lumen. At a dose of 0.6 mg/kg of nebulized nSta, the manifestations of ALI were completely eliminated. A significant dose-dependent reduction of the fibrin-positive areas in the lungs of mice with ALI was established. Nebulized nSta had a normalizing effect on the proinflammatory cytokines in blood— interleukin (IL)-1α, IL-17A, IL-6, and granulocyte-macrophage colony-stimulating factor (GM-CSF). These data showed the effectiveness of nebulized nSta and the perspectives of its clinical usage in COVID-19 patients with acute respiratory distress syndrome (ARDS).

## 1. Introduction

Acute respiratory distress syndrome (ARDS) is one of the most frequent complications of COVID-19, which is characterized by disrupture of the alveolar–capillary barrier with increased permeability, development of non-hydrostatic pulmonary edema, inflammation, and the appearance of anticoagulant factors in the lung [1]. These factors lead to the activation of pulmonary macrophages and an increased influx of neutrophils and platelets [2]. Complex coagulation and hematologic abnormalities, including significantly elevated D-dimer and fibrin/fibrinogen values, are the distinct features of a severe ARDS in COVID-19 patients [3]. A decrease in pulmonary fibrinolysis due to increased production of plasminogen activator inhibitor (PAI)-1, the main inhibitor of plasminogen activator in the lungs, was also found [4]. Along with increased levels of D-dimer and fibrinogen, ARDS is associated with a high pro-inflammatory status, cytokine “storm”, and macrophage and endothelial activation, with increased levels of interleukin (IL)-1, IL-6, IL-8, tumor necrosis factor (TNF)-alpha, ferritin, and C-reactive protein (CRP) [5]. Taken together, these factors result in the formation of fibrin-platelet microthrombi in the pulmonary vessels with the rapid development of progressive respiratory dysfunction [6].

The pathophysiology of ARDS involves the exudative stage, the development of thrombosis, the formation of a fibrin in the microcirculation, and the proliferation of connective tissue [7]. Theoretical considerations suggest that the usage of fibrinolytics may provide the effectiveness of ARDS treatment. Fibrinolytic therapy is also included in the list of antithrombotic drugs together with anticoagulant and antiplatelet therapy in patients with COVID-19 published by the Liverpool Drug Interactions Group [8]. However, in experimental models and in patients with ARDS, the positive effects of systemic anticoagulants and fibrinolytics were outweighed by systemic bleeding [9]. It seems promising that local administration of nebulized anticoagulants and fibrinolytics will improve pulmonary coagulopathy in ARDS and reduce the risk of systemic bleeding [10].

Non-immunogenic staphylokinase (nSta) (Fortelyzin^®^, INN/chemical name—a recombinant protein containing the amino acid sequence of staphylokinase, SuperGene, Moscow, Russia) is a high-fibrinselective thrombolytic agent with reduced production of antibodies and high biological activity. It is a single-chain molecule consisting of 138 amino acids with a molecular weight of 15.5 kDa. Due to replacement of amino acids in the immunogenic epitop of the Sta molecule, it lacks immunogenic properties. When nSta is added to human plasma containing a fibrin clot, it preferentially reacts with plasmin at the clot surface, forming a plasmin-nSta complex. This complex activates plasminogen trapped in the clot. The plasmin-nSta complex and fibrin-bound plasmin are protected from inhibition by α2-antiplasmin. Once liberated from the clot (or generated in plasma), however, they are rapidly inhibited by α2-antiplasmin. This selectivity of action confines the process of plasminogen activation to the clot, preventing excessive plasmin generation, α2-antiplasmin depletion, and fibrinogen degradation in plasma [11]. After nSta intravenous (i.v.) injection, the fibrinogen blood level decreases by less than 10% within the first 24 h. Thanks to the fact that nSta interacts only with a clot’s fibrin and does not interact with circulating fibrinogen, dose adjustment depending on body weight is not required. nSta was registered in Russia in 2012 as a thrombolytic drug for the treatment of patients with ST-segment elevation myocardial infarction (STEMI) [12] and in 2020 for the treatment of acute ischemic stroke [13].

In a multicenter, randomized clinical trial in patients with STEMI (FRIDOM), nSta was administered as a single i.v. bolus of 15 mg in all patients, regardless of bodyweight, and showed similarly high reperfusion patency and fewer minor bleeding events compared with tenecteplase, as well as the absence of neutralizing IgGs [14]. Results of the multicenter, randomized clinical trial in patients with an acute ischemic stroke (FRIDA) suggested that nSta administrated as a single i.v. bolus of 10 mg in all patients within the 4–5 h after the onset of symptoms is non-inferior to alteplase. Mortality, symptomatic intracranial hemorrhage, and serious adverse events did not differ between treatment groups [13].

Based on the role of pulmonary coagulation abnormalities in ARDS and the high fibrinolytic activity of nSta, we hypothesized that the nebulized nSta administration may be useful for ARDS therapy in COVID-19 patients. So, the aim of the study was to perform the toxicological study of nebulized nSta administration in healthy rats and the preclinical trial of the efficacy of nebulized nSta in a mouse model of acute lung injury (ALI) as a model of ARDS.

## 2. Results

### 2.1. Effect of Inhaled nSta on Coagulation Parameters

After 7 days of nSta nebulization in rats at a therapeutic dose of 0.2 mg/kg, as well as at doses 5 and 10 times higher than the therapeutic one (1.0 mg/kg and 2.0 mg/kg, respectively), no differences were found between the group of animals treated with saline (Table 1). Thus, inhaled nSta does not affect the fibrinogen blood level, activated partial thromboplastin time (APTT), and prothrombin time.

### 2.2. Mice ALI Model and Morphological Examination

After inhaled lipopolysaccharide (LPS) administration in the ALI non-treated group, 4 out of 10 animals died (Figure 1). In the control and all ALI groups treated with nebulized nSta, survival was 100%. During the 8 days of the experiment, untreated ALI animals gained significantly less weight compared to healthy animals, as well as animals treated with nSta (Table 2).

No changes were found in the lungs of the control animals that received nSta or saline solution by inhalation. The lungs were airy, without seals to the touch, and pale pink in color. There were patches of plethora. Macroscopic examination of the experimental groups ALI and ALI + nSta revealed damage in the lung tissue of various degrees, the most pronounced being in the ALI group (Table 3). The lungs had brown color with small patches of pale pink color retained, without compaction to the touch. There was a statistically significant increase in the lungs’ weight in the ALI group compared with the control intact group and the Control + nSta group and the normalization of this parameter in the ALI + nSta 0.2 mg/kg, ALI + nSta 0.4 mg/kg, and ALI + nSta 0.6 mg/kg groups (Table 3).

### 2.3. Histological Examination

Nebulized nSta or 0.9% sodium chloride solution in the control animals did not affect the lung structure. Bronchioles and bronchi were lined with cubic and cylindrical epithelium, and the lumens were clear (Figure 2A,B). The development of ALI in the lungs on the 8th day was accompanied by moderate edema. Diffuse infiltration around the bronchi with inflammatory elements was observed (Figure 2C). A decrease in the airiness of the lung tissue was noted, which was revealed in the form of incomplete collapse of the lung; the alveoli, as a result, had a lumen of irregular shape and different size. The interalveolar septa were thickened due to edema of the alveolar septa stroma. In a part of the alveoli, there were accumulations of exudate with an admixture of erythrocytes, fibrin, and desquamated alveolar epithelium.

Nebulized nSta at a dose of 0.2 mg/kg reduced ALI severity in animals on the 8th day after its administration. A decrease in the area of infiltration (3) and accumulation of exudate (4) was noted. The bronchi and small bronchioles had an open, clean lumen (2). Only small areas of lung tissue with unevenly thickened alveolar walls were observed. The lumen of large vessels was open; a moderate amount of blood was determined in the alveoli (Figure 2D). When nebulized nSta was used at a dose of 0.4 mg/kg, the animals showed no peribronchial edema, and the bronchi and small bronchioles had an open clear lumen. However, areas with thickened walls of the alveoli were noted (Figure 2E). Nebulized nSta at a dose of 0.6 mg/kg led to the elimination of ALI manifestations in animals. There were no areas of infiltration or accumulation of exudate and the bronchi and small bronchioles had an open clear lumen. Most of the alveoli were straightened and had an open clear lumen and thin walls. Only small areas with unevenly thickened alveolar walls were noted (Figure 2F).

### 2.4. Assessment of Fibrin/Fibrinogen Deposition in the Lungs

Fibrinogen-positive areas were detected in all ALI groups, as well as in mice of the control intact group (Figure 3B and Figure 4). In control mice treated with nebulized nSta (Figure 3A), in the absence of visible structural changes in the lung tissue, an insignificant increase in the fibrinogen content was revealed as compared to control mice treated with the saline.

The highest percentage of fibrinogen expression, along with pronounced tissue edema and thickening of the interalveolar septa, was observed in the ALI group—29% (Figure 3C). After 8 days of experiment, a significant dose-dependent decrease in the area of fibrinogen depositions in the lungs was found (Figure 3D–F): the proportion of fibrinogen-positive lung area in ALI animals treated with 0.2 mg/kg nSta was 21%; 0.4 mg/kg nSta—15%; 0.6 mg/kg—11% (Figure 4). An increase in nebulized nSta concentration by 0.2 mg/kg leads not only to a decrease in the percentage of fibrinogen-positive areas by 1.4–1.5 times, but also to a decrease in edema and expansion of the alveoli (Figure 3D–F).

### 2.5. Evaluation of the Cytokine Profile by Flow Cytometry

During the analysis, the following cytokines were determined in blood serum: IL-1α, IL-6, IL-17A, and granulocyte-macrophage colony-stimulating factor (GM-CSF) (Figure 5A–D, respectively).

Thus, on the 8th day of experiment, a statistically significant decrease in IL-1α, IL-17A, and GM-CSF levels was detected, and a twofold increase in the IL-6 level. Nebulization of nSta restored the changed parameters to the level of the control values. Statistically significant dose-dependent differences in the nSta influence on the cytokines profile were not found.

## 3. Discussion

In the first stage of our study we assessed the safety of nebulized nSta in healthy rats. It was found that nebulized nSta in a dose 10 times higher than the therapeutic one does not affect the fibrinogen blood level, APTT, and prothrombin time.

One of the most commonly used methods for ALI modeling is the usage of the *E. coli* bacterial endotoxin LPS [15]. In our study, in animals receiving inhaled LPS, the body weight gain was significantly less than in animals treated with nSta, which was combined with an increase in lung weight in comparison with healthy animals. Inhaled LPS administration caused pronounced edema of the connective tissue, diffuse infiltration by inflammatory cells, and a decrease in the airiness of the lung tissue. In a part of the alveoli, accumulations of exudate with an admixture of erythrocytes, fibrin, and desquamated alveolar epithelium were observed. Fibrinogen expression significantly rose, also accompanied with a significant increase in the fibrin-positive lung area in comparison with healthy animals. Inhaled LPS administration induced a systemic inflammatory response, as evidenced by a change in the proinflammatory cytokine profile. Thus, the inhaled LPS model did reproduce the main features of the classic ALI model in animals [16].

To the best of our knowledge, we are the first to establish the efficacy of nebulized nSta in the ALI model in mice. Nebulized nSta attenuated LPS-induced lung damage and did not affect the morphological structure of the lungs in healthy animals. A dose-dependent reduction in the fibrin-positive areas in the lungs without a change in the fibrinogen blood level and an absence of intra-alveolar hemorrhages were observed. Nebulized nSta had a normalizing effect on the concentration of the proinflammatory cytokines in blood—IL-1α, IL-17A, IL-6, and GM-CSF.

Mortality from ARDS has continued to be extremely high in recent years, especially during the COVID-19 pandemic. The pathogenesis of ARDS involves both proinflammatory and procoagulant mediators; the breakdown of the epithelial and endothelial barrier results in pulmonary edema, infiltration of neutrophils in the alveolar space, and deposition of intravascular and extracellular fibrin in the air spaces [15,16]. Along with the development of pulmonary edema and cytokine/chemokine storm, the most important aspect in ARDS pathogenesis is a direct link between inflammation and coagulation and a decrease in fibrinolytic activity [15,17,18,19,20]. The decreased fibrinolytic activity is predominately attributed to elevated PAI-1 both in the plasma and bronchoalveolar lavage fluid. The effect of nebulized nSta on the proinflammatory cytokines profile made a significant indirect contribution to preventing the development of coagulopathy together with the direct fibrinolysis in alveoli. Thus, fibrinolytic therapy may be potent for reducing neutrophil infiltration as well [21].

In a meta-analysis of fibrinolytics in preclinical trials there were reported to be at least 22 studies where fibrinolytic agents (tissue plasminogen activator, urokinase plasminogen activator, and plasmin) were administered by inhalation or intratracheally [21]. According to these preclinical data, the advantage of inhaled fibrinolytic therapy for ARDS in comparison with its i.v. administration could have a low risk of side effects, predominantly hemorrhages. It is well established in clinical trials that i.v. administration of fibrinolytics has numerous limitations and contraindications [22,23,24]. The inhalation of fibrinolytics may decrease these side effects in former trials. In our study, we proved that the usage of nebulized nSta allows thrombolytic treatment to be targeted in the lungs without serious adverse events, such as systemic bleeding.

For the moment, the results of the administration of inhaled fibrinolytics in several randomized clinical trials in COVID-19 patients with ARDS, which are ongoing (NCT02315898, NCT04356833), are not published. The results of a randomized clinical trial of inhaled streptokinase at a single dose of 250,000 IU for 4 h or 1,000,000 IU for 16 h in 40 patients with severe ARDS of non-COVID-19 etiology have been published. This trial showed that inhaled streptokinase, compared with heparin or standard therapy, statistically significantly reduced the number of deaths and improved blood oxygenation and hemodynamics [25].

Our study has some limitations. First, inhaled LPS administration in mice is a common ALI model that mimics human ARDS only in part, because this model cannot reflect the heterogeneous etiology and pathogenesis of ARDS [18]. Our experimental model focused only on the acute phase of lung injury. Second, the mice may have ingested some of the medication during nSta nebulization. Despite the limitations of our study, the obtained results correspond with previous investigations of inhaled fibrinolytics and anticoagulants in ALI models [2,26].

In conclusion, our experiments indicate that fibrinolytic therapy with nebulized nSta may diminish ALI severity by increasing fibrinolytic activity, suppressing lung injury, and normalizing proinflammatory cytokine levels. This allowed us to develop a protocol for a randomized clinical trial of nebulized nSta vs. placebo in COVID-19 patients with ARDS (FORRIF). The FORRIF trial protocol was registered on the ClinicalTrials.gov website (NCT05135546) and was approved by the Russian Ministry of Health (No. 733 10.11.2021) and the Ethics committee of the Russian Ministry of Health (No. 288 28.09.2021).

## 4. Material and Methods

### 4.1. Animals

Forty outbred male and female rats and sixty male C57BL/6 mice were purchased from the Federal State Budgetary Institution of Science’s Scientific Center for Biomedical Technologies of the Federal Medical and Biological Agency (Andreevka, Russia). Animals were individually housed in stainless steel cages in an air-conditioned room (22 ± 1 °C, 55 ± 5% humidity) with a 12 h/12 h light/dark cycle and *ad libitum* access to food (PK-120-1, Laboratorsnab, Moscow, Russia) and water. The study was conducted according to the guidelines of the Declaration of Helsinki and approved by the Bioethical Committee on Animal Health and Care of Privolzhsky Research Medical University in Nizhny Novgorod, protocol No. 14 of 24.09.2020.

### 4.2. Evaluation of Coagulation Parameters in Healthy Rats

The study was carried out on outbred male and female rats (1:1). Animals were randomly assigned into 4 experimental groups (Figure 6):Control intact (*n* = 10): saline nebulization;nSta 0.2 mg/kg (*n* = 10): nSta nebulization at a dose of 0.2 mg/kg;nSta 1.0 mg/kg (*n* = 10): nSta nebulization at a dose of 1.0 mg/kg;nSta 2.0 mg/kg (*n* = 10): nSta nebulization at a dose of 2.0 mg/kg.

Dosages of nSta were chosen on the basis of its efficacy in previous clinical trials in STEMI when it was i.v. administered. The maximum therapeutic dose of nSta for i.v. administration is 15 mg/day, which is equivalent to 0.2 mg/kg for a 75 kg patient. The interspecies dose transfer in mg/m^2^ was performed in accordance with the FDA recommendations, since nSta has a molecular weight of less than 100 kDa (15.5 kDa) [27]. To assess the effect of inhaled nSta on coagulation parameters, a therapeutic dose of 0.2 mg/kg was used, as well as five-fold (1.0 mg/kg) and ten-fold (2.0 mg/kg) increased dosages.

Nebulized nSta was administered daily for 7 days. The study of the coagulogram included the measurement of the fibrinogen blood level, APTT, and prothrombin time on days 7 and 21 of the experiment according to standard methods using programmable analyzer of hemostasis indicators APG2-02 (EMCO, Moscow, Russia). On day 22, the animals were euthanized by placement in a CO_2_ chamber for 10–20 s.

### 4.3. Acute Lung Injury (ALI) Modeling

The ALI model in mice was induced in 40 mice by inhalation of 10 mg/kg LPS (Escherichia coli O111: B4, Sigma, St. Louis, MO, USA), and 20 mice were used as controls (10 received saline, 10 received nSta) (Figure 7) [28]. The mortality rate in this model is 40–60%. In the experiment, an inhalation system of an original design was used, which allows simultaneously accommodating 1 or up to 10 small laboratory animals. The system recognizes oneself as a sealed chamber for animals with a nebulizer, inlet pipes into the chamber, and outlet pipes from the chamber, equipped with special degassing traps [29]. The LPS simulating the ALI dose was 10 mg/kg with an air supply of 2 L/h per animal with an exposure of 90 min (0.16 L/min).

A 0.9% sodium chloride solution (Avexima, Moscow, Russia) or nSta were nebulized at doses of 0.2 mg/kg, 0.4 mg/kg, or 0.6 mg/kg 30 min, 24 h, or 48 h after LPS inhalation. Two- and three-fold increasing of the nSta therapeutic dose were chosen for inhalation, taking into account the lower bioavailability and losses during inhalation and a possible dose-dependent efficacy.

### 4.4. ALI Experimental Groups

Animals were randomly assigned into 6 experimental groups:Control nSta (*n* = 10): nSta nebulization at a dose of 0.2 mg/kg;Control intact (*n* = 10): saline nebulization;ALI (*n* = 10): saline nebulization;nSta (*n* = 10): nSta nebulization at a dose of 0.2 mg/kg;nSta (*n* = 10): nSta nebulization at a dose of 0.4 mg/kg;nSta (*n* = 10): nSta nebulization at a dose of 0.6 mg/kg.

### 4.5. Necropsy

On day 8, the surviving animals were euthanized by placing them in a CO_2_ chamber, followed by necropsy for macroscopic examination. Necropsy was performed under the direct supervision of a pathologist. The lungs were weighed; their histological study was performed.

### 4.6. Histological Examination

Lungs were fixed in 10% neutral formalin, dehydrated in alcohols of ascending concentrations and xylene, and embedded in paraffin. Paraffin sections 5 µm thick were obtained using SM 2000R microtome (Leica, Wetzlar, Germany), stained with hematoxylin and eosin (H&E), and examined using a DM1000 microscope (Leica, Wetzlar, Germany).

The grading of the infiltration of exudate in the alveoli was performed as follows: 1—no infiltration; 2—weak (10% of the lung area); 3—moderate (25% of the lung area); 4—intermediate (50% of the lung area); 5—severe (>50% of the lung area).

The grading of the stroma edema and thickening of the alveolar septa was performed as follows: 1—no changes; 2—moderate (<25% of the lung area); 3—intermediate (25–50% of the lung area); 4—severe (>50% of the lung area).

### 4.7. Assessment of Fibrin/Fibrinogen Deposits in the Lungs

For immunohistochemical staining, 5 µm paraffin sections of lung tissue were deparaffinized using xylene and 96% and 70% alcohol and boiled for 20 min in a universal antigen retrieval reagent (ab208572, Abcam, Cambridge, UK) in a microwave oven. A hydrogen peroxide blocking solution (ab64218, Abcam, Cambridge, UK) was applied to the entire tissue section surface for 10 min. Protein blocking solution (ab64226, Abcam, Cambridge, UK) was used for 5 min at room temperature to reduce nonspecific background staining.

Sections were stained with primary rabbit polyclonal antibodies against fibrinogen (ab34269, Abcam, Cambridge, UK) in a solution of 1% BSA in PBS overnight at 4 °C. A rabbit-specific IHC polymer detection kit HRP/DAB (ab209101, Abcam, Cambridge, UK), was used for signal amplification. Tissue sections were stained with a DAB chromogen (ab64238, Abcam, Cambridge, UK) for 2 min, and the cell nuclei were stained with hematoxylin (ab220365, Abcam, Cambridge, UK) for 5 min at room temperature. After every step, the sections were washed with TBS solution (ab64204, Abcam, Cambridge, UK) 3–4 times. The preparations were finally mounted under cover glasses using a mounting medium for IHC (ab64230, Abcam, Cambridge, UK).

Light images were registered with a Axio Scope A1 microscope (Zeiss, Oberkochen, Germany) using 20× dry objective and ZEN 3.0 software (Zeiss, Oberkochen, Germany). Fibrinogen expression in the obtained images were analyzed using the ImageJ Fiji software (version 1.2; WS Rasband, National Institute of Health, Bethesda, Rockville, MD, USA) based on the previously described protocol [30].

The percentage ratio of fibrinogen-positive areas to the total area of lung tissue was calculated by the following equation:% fibrinogen = (S_fibrinogen_/S_lungs_) × 100%(1)
where % of fibrinogen was the percentage of fibrinogen expression regions,

S_fibrinogen_ is the area of the fibrinogen-positive regions in the image, and

S_lungs_ is the area of the lung tissue in the image.

### 4.8. Evaluation of the Cytokine Profile by Flow Cytometry

Blood was collected from the tail vein on day 8 of the experiment. Samples were centrifuged at 3000× *g* for 15 min. IL-1α, IL-6, IL-17A, and GM-CSF were determined in serum by multiplex assays by a BD FACS CantoII flow cytometer (Becton Dickinson, Bergen County, NJ, USA). A LEGENDplexTM Mouse Inflammation panel kit (Cat. No. 740150, BioLegend Way, San Diego, CA, USA) was used according to the manufacturer’s instructions. Then, 25 µL of samples or standard were added per well, 25 µL of the Capture Beads solution was added, and they were incubated for 2 h at room temperature. After washing, 25 µL biotinylated antibodies were added to each well and incubated for 1 h at room temperature. Then, 25 μL of streptavidin-phycoerythrin solution was added and incubated for 30 min at room temperature. At the end of the incubation, the samples were washed, and 150 μL of phosphate buffer was added to each well. Samples were analyzed on a flow cytometer at 575 nm and 660 nm. The data were processed using the LEGENDplexTM Data Analysis Software version 7.1 (BioLegend Way, San Diego, CA, USA).

### 4.9. Statistical Analysis

Results are expressed as the mean and standard deviation (SD) or standard error of the mean (SEM). The data were checked for normal distribution using the Anderson–Darling test. Nonparametric tests were used to compare differences among small samples (*n* ≤ 10), the Mann–Whitney U test was used to compare differences between two independent samples (control vs. experimental), and the Kruskal–Wallis test with Dunn’s post hoc analysis was used to compare differences among more than two samples. The reliability of immunohistochemical data was assessed using Brown–Forsythe, an ANOVA test, and the Games–Howell test. Differences between groups were considered statistically significant at *p* < 0.05. All statistical analyses were performed using GraphPad Prism software version 9.2.0 (GraphPad Software, San Diego, CA, USA).

## Figures and Tables

**Figure 1 ijms-23-09307-f001:**
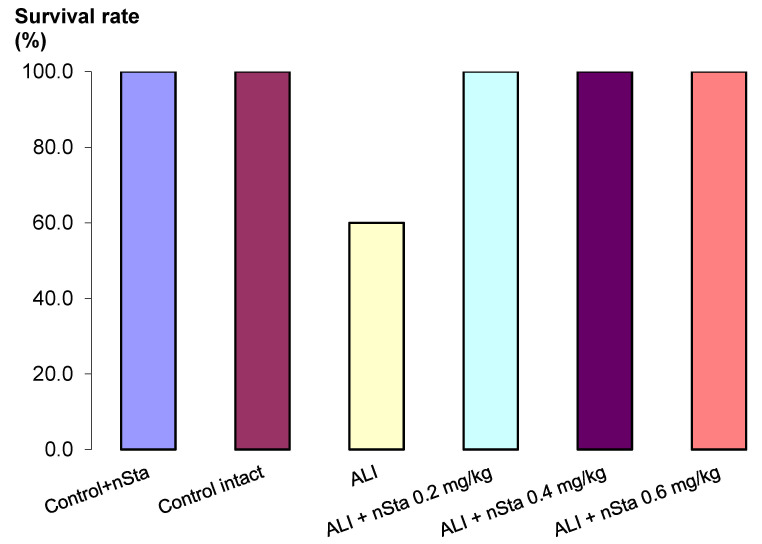
Survival analysis of Control + nSta, Control intact, untreated lipopolysaccharide (LPS)-induced acute lung injury (ALI) mice, and LPS-induced ALI mice receiving nSta 0.2 mg/kg, nSta 0.4 mg/kg, or nSta 0.6 mg/kg for up to 8 days after LPS administration (*n* = 10 for each group at the beginning of the experiment).

**Figure 2 ijms-23-09307-f002:**
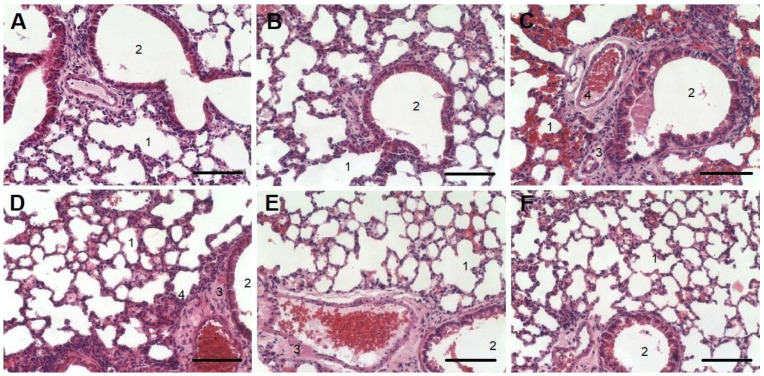
Histology examination of lung tissue for the Control + nSta (**A**), Control intact (**B**), and ALI (**C**) groups and for ALI groups treated with nSta inhalation ((**D**)—0.2 mg/kg, (**E**)—0.4 mg/kg, (**F**)—0.6 mg/kg) stained with hematoxylin and eosin. Magnification × 20. Scale—100 μm. 1—alveoli; 2—bronchial lumen; 3—areas of infiltration and alveolar decline; 4—accumulation of exudate.

**Figure 3 ijms-23-09307-f003:**
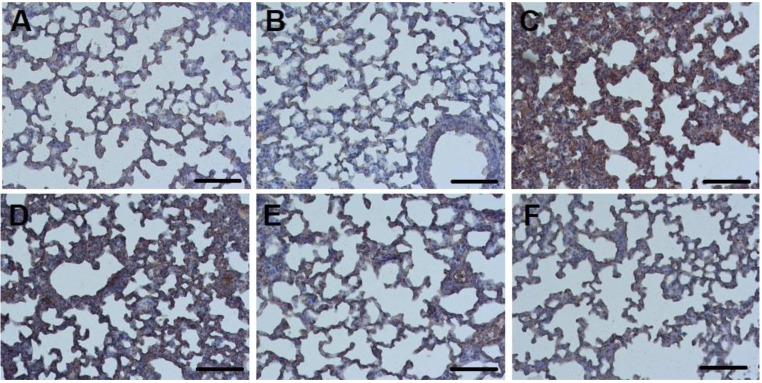
Sections of lung tissue in the Control + nSta (**A**), Control intact (**B**), and ALI groups (**C**) and in the ALI groups treatment with nSta inhalation ((**D**)—0.2 mg/kg, (**E**)—0.4 mg/kg, (**F**)—0.6 mg/kg), stained with anti-fibrinogen antibodies. Cell nuclei are stained with hematoxylin. Magnification × 20. Scale—100 μm.

**Figure 4 ijms-23-09307-f004:**
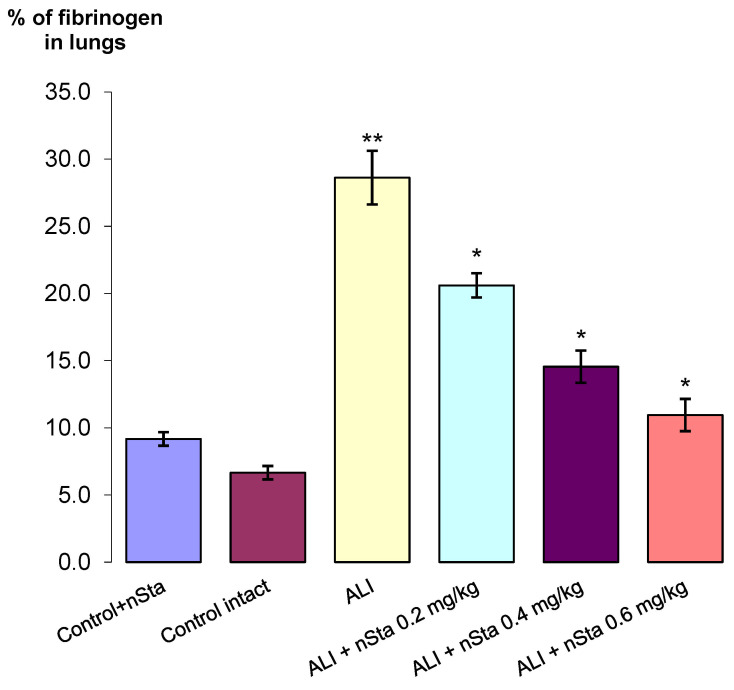
The percentage of fibrinogen-positive areas in the total area of lung tissue in the control, nontreated, and ALI groups after nSta 0.2 mg/kg, nSta 0.4 mg/kg, and nSta 0.6 mg/kg nebulization. Data are presented as the M ± SD. * *p* < 0.05 for ALI + nSta 0.2 mg/kg vs. ALI, ALI + nSta 0.4 mg/kg vs. ALI, ALI + nSta 0.6 mg/kg vs. ALI; ** *p* < 0.01 for ALI vs. Control Intact (ANOVA test with the Games–Howell post hoc test).

**Figure 5 ijms-23-09307-f005:**
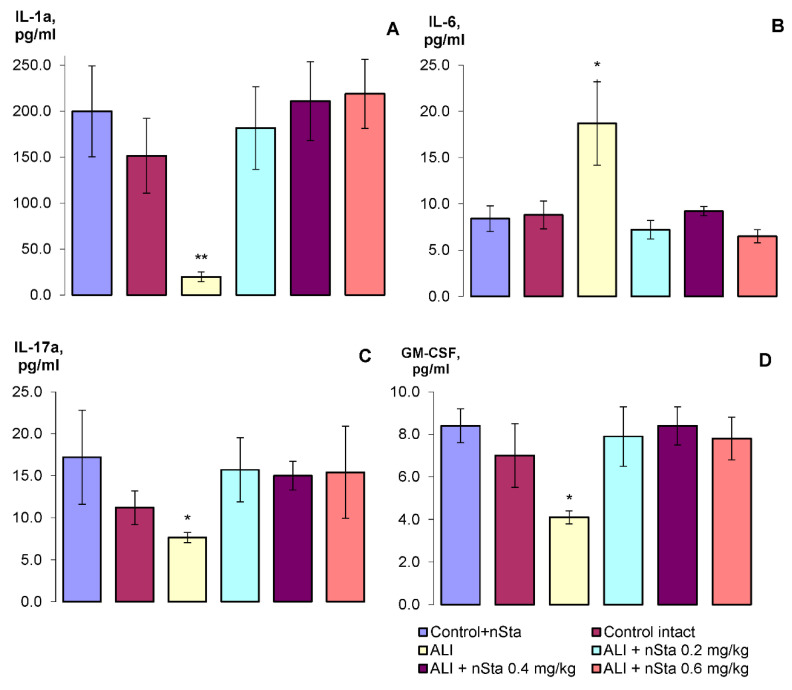
The profile of the cytokines of the experimental animals: (**A**)—IL-1α, (**B**)—IL-6, (**C**)—IL-17A, (**D**)—GM-CSF. Data are presented as the M ± SD. * *p* < 0.05, ** *p* < 0.01 for ALI vs. Control intact (ANOVA test with the Games–Howell post hoc test).

**Figure 6 ijms-23-09307-f006:**
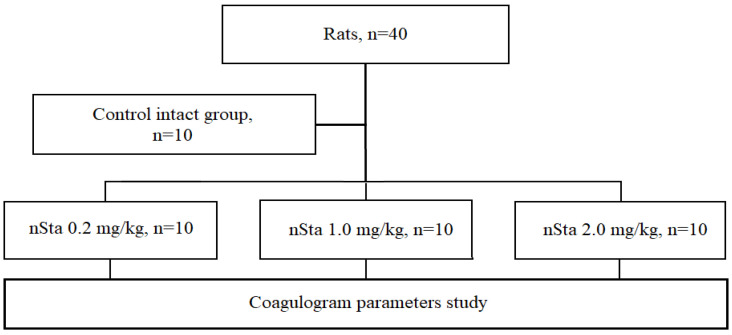
Flow chart of the first part of the study: evaluation of nSta on the coagulogram parameters in healthy rats.

**Figure 7 ijms-23-09307-f007:**
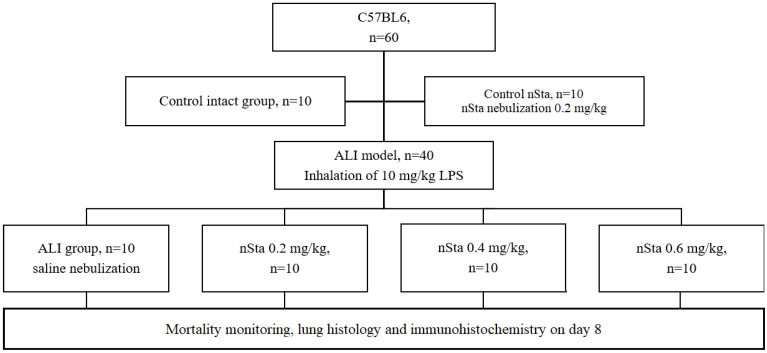
Flow chart of the second part of the study: evaluation of the effects of nSta on the mice ALI model.

**Table 1 ijms-23-09307-t001:** Effect of inhaled nSta on coagulation parameters.

Groups	Fibrinogen, g/L (M ± SD)		APTT, s (M ± SD)		Prothrombin Time, s (M ± SD)	
	Day 7	Day 21	Day 7	Day 21	Day 7	Day 21
Control intact (*n* = 10)	1.96 ± 0.10	1.91 ± 0.13	15.26 ± 062	15.36 ± 0.63	22.44 ± 0.85	22.18 ± 0.58
nSta 0.2 mg/kg (*n* = 10)	1.91 ± 0.09	1.93 ± 0.09	15.14 ± 0.65	15.40 ± 0.46	22.34 ± 0.74	22.12 ± 0.45
nSta 1.0 mg/kg (*n* = 10)	1.93 ± 0.08	1.96 ± 0.09	15.24 ± 0.34	15.18 ± 0.55	22.20 ± 0.83	22.16 ± 0.71
nSta 2.0 mg/kg (*n* = 10)	1.92 ± 0.10	1.94 ± 0.13	15.32 ± 0.66	15.12 ± 0.56	22.20 ± 0.80	22.24 ± 0.83

**Table 2 ijms-23-09307-t002:** Body weights of experimental animals.

Groups	Weight before ALI Induction, g (M ± SD)	Weight on Day 8, g (M ± SD)
Control nSta (*n* = 10)	19.56 ± 0.14	26.31 ± 0.95
Control intact (*n* = 10)	18.44 ± 0.75	27.12 ± 0.63
ALI (*n* = 6)	19.87 ± 0.25	21.20 ± 0.75 *
ALI + nSta 0.2 mg/kg (*n* = 10)	19.26 ± 0.32	23.54 ± 0.24
ALI + nSta 0.4 mg/kg (*n* = 10)	18.63 ± 0.27	25.37 ± 0.59
ALI + nSta 0.6 mg/kg (*n* = 10)	18.92 ± 0.31	26.63 ± 0.10

* *p* < 0.01 for ALI vs. Control intact (Kruskal–Wallis test with Dunn’s post hoc test).

**Table 3 ijms-23-09307-t003:** Weights and morphological characteristics of the lungs of experimental animals.

Groups	Lungs’ Weight, g (M ± SD)	Exudate Score in Lungs (M ± SD)	Interstitial Edema and Hyaline Membrane Formation Score in Lungs (M ± SD)
Control nSta (*n* = 10)	0.165 ± 0.011	0	1
Control intact (*n* = 10)	0.161 ± 0.028	0	1
ALI (*n* = 6)	0.233 ± 0.013 *	2.33 ±0.51 **	3.66 ± 0.51 **
ALI + nSta 0.2 mg/kg (*n* = 10)	0.191 ± 0.014 ^	1.1 ± 0.73	2.2 ± 0.78
ALI + nSta 0.4 mg/kg (*n* = 10)	0.187 ± 0.024 ^	0.6 ± 0.51 ^^	1.4 ± 0.51 ^
ALI + nSta 0.6 mg/kg (*n* = 10)	0.161 ± 0.008 ^^	0.5 ± 0.52 ^^	1 ^^

* *p* < 0.05, ** *p* < 0.01 for ALI vs. Control intact; ^ *p* < 0.05, ^^ *p* < 0.01 for ALI + nSta vs. ALI (Kruskal–Wallis test with Dunn’s post hoc test).

## Data Availability

Additional data can be provided upon reasonable request from the date of publication of this article within 5 years. The request should be sent to the corresponding author at gendirector@supergene.ru.

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
