# Peer review of "Nebulized Non-Immunogenic Staphylokinase in the Mice Acute Lung Injury Model"

_ijms, 2022, doi:10.3390/ijms23169307_

Round 1

Reviewer 1 Report

Will the LPS-induced ARDS be similar to a viral infected (COVID) immune phenotype in the Lung? The authors mentioned that COVID leads to activated macrophages, the influx of neutrophils, and high platelets. Did the authors determine the impact of nSta on immune cell function? What mechanisms restore the cytokine storm in ALI? [Data Table 3, IL-6]

Why didn't the authors measure the lung microenvironment/cytokine profile using BAL instead of the blood? There is COVID literature that measured the lung cytokines. [For example, https://doi.org/10.3389%2Ffimmu.2020.621441]

Can the authors provide the standard curve for the FC Multiplex assay?

Can the authors provide the profile of all the 13 cytokines in the Kit/ assay from BioLegend? 

Figure 4, typo Lung.

Author Response

Dear Reviewer, we are grateful for careful reading and analysis of our manuscript, and questions. Below are our responses and comments.

Will the LPS-induced ARDS be similar to a viral infected (COVID) immune phenotype in the Lung? The authors mentioned that COVID leads to activated macrophages, the influx of neutrophils, and high platelets. Did the authors determine the impact of nSta on immune cell function? What mechanisms restore the cytokine storm in ALI? [Data Table 3, IL-6].

Since nSta is a fibrinolytic agent, the determining criterion for choosing a model for evaluating its effectiveness for nebulization was a violation of hemocoagulation and fibrin deposition in the lungs. Inhaled LPS administration is known to have caused pronounced edema of the connective tissue and diffuse infiltration by inflammatory cells. Fibrinogen expression significantly raised, also accompanied with a significant increase in the fibrin-positive lung area in comparison with healthy animals. In addition, inhaled LPS administration induced a systemic inflammatory response, as evidenced by a change in the proinflammatory cytokine profile, in particular IL-6, IL-1a, TNF and others. In this respect immune phenotype of LPS-induced ARDS is similar to that of COVID [Huang C., et al., 2020, Lancet; Matsuyama Т., et al., 2020, Cell Death Differ.]. The effect of nSta on immune cell function was not within the scope of this study.

Why didn't the authors measure the lung microenvironment/cytokine profile using BAL instead of the blood? There is COVID literature that measured the lung cytokines. [For example, https://doi.org/10.3389%2Ffimmu.2020.621441]

As shown before in “Meta-Analysis of Preclinical Studies of Fibrinolytic Therapy for Acute Lung Injury” [Liu C., 2018, Front. Immunol.], effects of fibrinolytic treatment on the fibrinolysis, PAI-1 level and neutrophils infiltration were similar both in the plasma and BALF. Based on this, we chose blood to evaluate the effectiveness of the nebulized nSta therapy, as it was done before [Kudinov V.A et al.., 2022, Int. J. Mol. Sci.].

Can the authors provide the standard curve for the FC Multiplex assay?

We attached the standard curves for LEGENDplexTM Mouse Inflammation panel kit (740150). The curves of our laboratory corresponded to those given by 97%.

Can the authors provide the profile of all the 13 cytokines in the Kit/ assay from BioLegend? 

We measured all the 13 cytokines in the Kit/ assay: IL-1β, IL-10, IL-12p70, IL-23, IL-27, CCL2(MCP-1), IFN-β, IFN-γ, TNF-α, IL-1α, IL-6, IL-17A, GM-CSF.

There were no changes in the following serum cytokine profile: IL-1β, IL-10, IL-12p70, IL-23, IL-27, CCL2(MCP-1), IFN-β, IFN-γ, TNF-α.

Data on changes in IL-1α, IL-6, IL-17A, GM-CSF cytokines blood level of the experimental animals are presented in Fig. 5.

Figure 4, typo Lung. - Сorrected.

Reviewer 2 Report

The current manuscript by Markin et al. describes a study on the efficacy of non-immunogenic recombinant staphylokinase (nSta) protein in ameliorating the symptoms in LPS-induced acute lung injury (ALI) in mice model.

Previously, the efficacy of nSta has already been demonstrated through clinical trials (Lancet Neurol. 2021 Sep;20(9):721-728 and RUDN Journal of Medicine. 2012 Issue: No 1 Pages: 105-110). The pre-clinical study on the effects of nSta treatment is reported here using two animal models. First authors used healthy rats to evaluate the blood coagulation parameters following different doses of nSta. Following the examination of the safety of nSta, authors used LPS-induced mice ALI model to evaluate the efficacy of nSta. Based on the observations of the current study, authors are performing a clinical trial of nebulized nSta in COVID-19 patients.

Authors are requested to address the following suggestions. Accordingly, authors may consider editing the limitations paragraph in the Discussion section.

Main suggestions

1. Authors are requested to provide more quantitative assessments of the lung injury caused by LPS, and the subsequent improvement by nSta treatment. Currently, in table 2, authors reported lung’s weight, exudate score, and interstitial edema and hyaline membrane formation score. Authors should consider including some additional histopathological parameters as well.

2. It is interesting to see that nSta treatment caused non-significant increase in fibrin/fibrinogen deposition in the lungs. Authors may consider examining whether this is changed if the animals are euthanized at a later time point than on the eighth day. If the rat lungs’ samples were fixed, authors may stain the sections for fibrin/fibrinogen deposition and check whether any sign of fibrin deposition is noted there.

3. The nSta treatment was done following LPS treatment. A question remains unanswered is whether pre-treatment with nSta before LPS treatment will have any beneficial effects or not. Authors are requested to comment on this.

4. Results, 2.2. Mice ALI model and morphological examination, authors reported that mice lungs had “brown color with small patches of pale pink color” following ALI induction. If possible, authors may consider including pictures of the lungs to support these observations.

5. From table 2, it is evident that only the highest dose of nSta is effective in normalizing the interstitial edema and hyaline membrane formation score. When compared between ALI (3.66) and ALI+0.2 mg/kg (2.2) groups, is the difference significant? Similarly, is the exudate score between ALI (2.33) and ALI+0.2 mg/kg (1.1) groups significantly different?

6. In figure 2C and 2D, authors identified areas where “infiltration and alveolar decline” and “accumulation of exudate” have occurred. It is very difficult to confidently identify these pathological alterations from these current histology fields. Authors are requested to provide more convincing images to prove this point.

7. The change in IL-1alpha in blood requires additional explanations. From table 3, it seems that at higher doses of nSta the concentration is increasing. This may indicate impact on inflammatory responses. Also, the overall change of this cytokine across the various groups is rather inconsistent.

8. Discussion, authors mentioned here that there was a decrease in the body weight of the animals receiving LPS. Authors are requested to report the body weights of animals under all of the groups and the effect of nSta treatment on this parameter.

9. At some point authors should consider estimating how much nSta is actually reaching the lungs.

Other comments

1. The manuscript requires thorough editing for language use.

2. The last paragraph of the introduction is rather confusing. Authors are requested to indicate the study plan in brief, and the rationale.

3. Methods, 4.2. Evaluation of coagulation parameters in healthy rats, please report the details of how the nSta was nebulized. Also, authors mentioned that on day 22 rats were euthanized by placing them in carbon dioxide chamber. Did authors perform any histological examination thereafter of the lungs?

4. Methods, 4.3. Acute lung injury (ALI) modeling, “ALI model in mice was induced in 60 mice by inhalation of 10 mg/kg LPS”; authors induced ALI in 40 out of 60 animals. Also, more clarification of the study protocol is needed. It seems that authors dosed the ALI animals three times following LPS treatment; after half an hour, one day and two days. And at the eighth day authors euthanized the animals. Please include all of the relevant details.

5. Please clarify whether blood serum or plasma was used for cytokine profiling.  And how this was isolated from the whole blood.

6. Please define the abbreviations at the first mention.

7. Current study was done on male animals, hence some discussion on the effect of sex will be useful.

Author Response

Dear Reviewer, we are grateful for careful reading and analysis of our manuscript, and questions. Below are our responses and comments.

  1. Authors are requested to provide more quantitative assessments of the lung injury caused by LPS, and the subsequent improvement by nSta treatment. Currently, in table 2, authors reported lung’s weight, exudate score, and interstitial edema and hyaline membrane formation score. Authors should consider including some additional histopathological parameters as well.

In “Meta-Analysis of Preclinical Studies of Fibrinolytic Therapy for Acute Lung Injury” [Liu C. et al., 2018, Front. Immunol.] lung injury scores are proposed to be analyzed with occurrence of at least one of following indices: (1) leukocyte and red blood cell infiltration, (2) alveolar epithelium damage, (3) interstitial edema, and (4) fibrin deposition and hyaline membrane formation. We did not count leukocyte and red blood cell infiltration since it is more informative in BALF. Nevertheless, the cytokines blood level was determined to assess the development of ALI. We did not count the alveolar epithelium damage. For assessments LPS-induced lung injury we choose the indicators such as lung’s weight, exudate score, interstitial edema and hyaline membrane formation, as well as fibrin deposition. These indicators sufficiently demonstrate the damaging effect of LPS in the lungs, confirm the development of ALI, and prove the dose-dependent therapeutic effect of nSta.

  1. It is interesting to see that nSta treatment caused non-significant increase in fibrin/fibrinogen deposition in the lungs. Authors may consider examining whether this is changed if the animals are euthanized at a later time point than on the eighth day. If the rat lungs’ samples were fixed, authors may stain the sections for fibrin/fibrinogen deposition and check whether any sign of fibrin deposition is noted there.

Fibrin/fibrinogen deposition in Control+nSta group was comparable to the Control group (fig. 4). In most (8 of 10) of the conducted preclinical studies of fibrinolytics in the ALI model determined death within 2 days. Longer observation of animals in two studies (2 days up to 30 days) did not affect the results of the studies [Liu C. et al., 2018, Front. Immunol.]. In our experiment all animals were euthanized on the 8th day of the experiment for pathological examination of the lungs.

  1. The nSta treatment was done following LPS treatment. A question remains unanswered is whether pre-treatment will have any beneficial effects or not. Authors are requested to comment on this.

As to mortality, preventive strategy showed a beneficial effect, whereas treatment strategy did not. Treatment strategy showed a beneficial effect on PaO2, but not preventive one [Liu C. et al., 2018, Front. Immunol.]. Since the nebulized nSta is planned to be used in clinical practice in patients with COVID-19, its preventive use will be difficult.

  1. Results, 2.2. Mice ALI model and morphological examination, authors reported that mice lungs had “brown color with small patches of pale pink color” following ALI induction. If possible, authors may consider including pictures of the lungs to support these observations

We have modified Figure 2C.

  1. From table 2, it is evident that only the highest dose of nSta is effective in normalizing the interstitial edema and hyaline membrane formation score. When compared between ALI (3.66) and ALI+0.2 mg/kg (2.2) groups, is the difference significant? Similarly, is the exudate score between ALI (2.33) and ALI+0.2 mg/kg (1.1) groups significantly different?

There are no significant differences between the ALI and nSta 0.2 mg/kg groups (p=0.138) in terms of Exudate score in lungs and between the ALI and nSta 0.2 mg/kg groups (p=0.210) in terms of Interstitial edema and hyaline membrane formation score in lungs.

The significantly differences become with the dosage of nSta 0.4 mg/kg (p=0.005 compared with ALI in terms of Exudate score and p=0.021 compared with ALI in terms of Interstitial edema) and nSta 0.6 mg/kg (p=0.003 and p=0.001 respectively).

  1. In figure 2C and 2D, authors identified areas where “infiltration and alveolar decline” and “accumulation of exudate” have occurred. It is very difficult to confidently identify these pathological alterations from these current histology fields. Authors are requested to provide more convincing images to prove this point

We have modified Figure 2C, it represents an increased infiltration (3), accumulations of exudate with an admixture of erythrocytes (4).

Figure 2D represents the lung of ALI animals treated with the minimum dose of nSta 0.2 mg/kg. There is no pronounced improvement in the morphological structure of the lungs at this dose, however, a decrease in the area of infiltration can be noted (3), as can be seen in the figure.  

  1. The change in IL-1alpha in blood requires additional explanations. From table 3, it seems that at higher doses of nSta the concentration is increasing. This may indicate impact on inflammatory responses. Also, the overall change of this cytokine across the various groups is rather inconsistent.

After LPS nebulization IL-1a blood level decreased by 10 times compared with the control groups. nSta nebulization dose-dependently restores the level of IL-1a to the values ​​of control animals.

  1. Discussion, authors mentioned here that there was a decrease in the body weight of the animals receiving LPS. Authors are requested to report the body weights of animals under all of the groups and the effect of nSta treatment on this parameter.

We have added a table 2 with the body weight of the animals to the article.

  1. At some point authors should consider estimating how much nSta is actually reaching the lungs.

According to the mentions above “Meta-Analysis of Preclinical Studies of Fibrinolytic Therapy for Acute Lung Injury” [Liu C., 2018, Front. Immunol.] “the best route for ALI treatment is intratracheal delivery”. Quantitative determination of the nebulized nSta bioavailability in the lungs has not been carried out. Delivery of drug to the airway mucosa by inhalation therapy depends on many factors, including the pattern of breathing, the geometry of lungs and airways (often altered in patients with lung disease), and the size of the aerosol particles. Approximately 30% of the delivered dose actually reaches the lung under optimal conditions [Melin J., et al., 2017, AAPS J.]. Based on these data, we proposed a threefold increase in the nSta therapeutic dose which showed the maximum efficiency according to the results of the experiments.

Other comments

  1. The manuscript requires thorough editing for language use.

We agree with the need to improve the language and are ready to use Language Editing Services offered by MDPI.

  1. The last paragraph of the introduction is rather confusing. Authors are requested to indicate the study plan in brief, and the rationale.

We have corrected the last paragraph of the introduction.

  1. Methods, 4.2. Evaluation of coagulation parameters in healthy rats, please report the details of how the nSta was nebulized. Also, authors mentioned that on day 22 rats were euthanized by placing them in carbon dioxide chamber. Did authors perform any histological examination thereafter of the lungs?

For nSta nebulization original system as described in 4.3 section was used. The system recognizes oneself as a sealed chamber for 5 rats with a nebulizer, inlet pipes into the chamber and outlet pipes from the chamber, equipped with special de-gassing traps.

The histology examination of lung tissue of the healthy rats 22 days after nSta nebulization was performed. There were no differences in lung structure between animals treated with nSta or saline.

  1. Methods, 4.3. Acute lung injury (ALI) modeling, “ALI model in mice was induced in 60 mice by inhalation of 10 mg/kg LPS”; authors induced ALI in 40 out of 60 animals. Also, more clarification of the study protocol is needed. It seems that authors dosed the ALI animals three times following LPS treatment; after half an hour, one day and two days. And at the eighth day authors euthanized the animals. Please include all of the relevant details.

This section was corrected.

  1. Please clarify whether blood serum or plasma was used for cytokine profiling. And how this was isolated from the whole blood.

For the study, blood was taken from the surviving animals on the 8th day of the experiment before euthanasia. Samples were centrifuged at 3000g for 15 minutes. Serum in a volume of 0.5 ml was stored at -70°C until the cytokine profile by flow cytometry.

  1. Please define the abbreviations at the first mention.

All abbreviations are defined.

  1. Current study was done on male animals, hence some discussion on the effect of sex will be useful.

Safety studies of the nebulized nSta were carried out on healthy rats, both male and female (1:1). No sex differences were found for any of the analyzed indicators. Therefore, efficacy studies have only been conducted on male mice.

Round 2

Reviewer 1 Report

Thanks to the authors for the comprehensive responses to my previous review. I was able to understand the data output and inferences based on the additional data that was included. Even though I would still prefer BAL since it's a direct measure of lung profile, the authors provided reference citations to support their design and choice to use blood instead of BAL. 

I appreciate the authors that they commented on my interest in the other unreported cytokines in their multiplex assay. Interesting that there was no IFNg or TNF induction. 

Good luck!

Reviewer 2 Report

Authors tried their best to address the comments.